# Modeling of the Chinese Dating App Use Motivation Scale According to Item Response Theory and Classical Test Theory

**DOI:** 10.3390/ijerph192113838

**Published:** 2022-10-24

**Authors:** Fen Ren, Kexin Wang

**Affiliations:** 1School of Education and Psychology, University of Jinan, Jinan 250022, China; 2College of Media and International Culture, Zhejiang University, Hangzhou 310058, China

**Keywords:** dating app, motivation, item response theory, CTT, Chinese version

## Abstract

Dating apps are popular worldwide among young adults, and the Tinder use motivation scale is widely used to measure the primary motives for dating app use. In light of the increasing prevalence of dating apps among young Chinese adults, this study applied both item response theory and traditional classical test theory to examine the psychometric properties of the Chinese version of the dating app use motivation scale that is applicable across different dating apps. In total, 1046 current or former dating app users (age range: 18–30, *M* = 26.20, *SD* = 4.26, 52.30% girls) completed the online survey. From the original item pool, this study selected 25 items according to item response theory analysis, retracted six factors based on exploratory factor analysis (EFA), and conducted confirmatory factor analysis for further validation. The motivations were seeking a relationship, self-worth validation, the thrill of excitement, ease of communication, emotion-focused coping, and fun. The first four motivations were consistent with the original scale, and two new motivations were found in the present sample. All six motivations were validated among the Chinese sample. Not consistent with the Tinder use motivation scale, casual sex was not identified as a primary motivation among young Chinese adults. One related measure was used to obtain convergent validity. The discussion focused on the cultural and methodological factors that may explain the differences between the original scale and the Chinese version of the scale.

## 1. Introduction

Dating apps enable people to pursue a romantic or sexual partner using their phone. The past decade has witnessed a dramatic growth in dating app use: Tinder has been downloaded more than 430 million times and led to over 60 billion matches across 190 countries since its launch in 2012 [1]. Amid the pandemic, the number of exchanges and the length of interactions on Tinder saw a sharp increase because many in-person social areas were shut down [2]. People from Western countries mainly use Tinder, while Chinese people use local apps with similar functions. Momo, the most popular local dating app, had over 115 million users in mainland China [3].

Research has shown that young adults are avid users, and they use Tinder due to a diverse set of motivations [4,5]. Such different motivations may affect their romantic and sexual life in either a positive or undesirable way. For instance, using the Tinder use motivation scale developed by Sumter et al., researchers found that casual sex motivation is associated with higher risk of sexual behavior (e.g., having unprotected sex) [5,6]. However, affection motivations for using dating apps are associated with long-term mating strategies [7].

Although the Tinder use motivation scale has been used in empirical studies across different dating app platforms and various cultures, its application for Chinese local dating app Momo is questionable due to several methodological issues. Therefore, we aim to reexamine the Tinder use motivation scale among Chinese emerging adults by addressing these issues. Understanding why young adults use dating apps would pave the way for organizing preventions that may decrease potential detrimental outcomes, and promote practices that may facilitate utility.

### 1.1. Measuring Tinder Use Motivation

The most commonly cited version of the dating app use motivation scale was developed by Sumter et al. and based on Tinder [5]. This motivation scale has been used in empirical studies among dating app users from the United States, the Netherlands, German, and India [8,9,10,11]. Sumter et al. selected 24 items, from an initial pool of 46 items, using exploratory factor analysis with a parallel analysis approach with a small sample size (n = 163), uncovering six motivations for using Tinder: love, casual sex, ease of communication, self-worth validation, the thrill of excitement, and trendiness [5]. 

Timmermans and De Caluwé developed the Tinder motives scale (TMS), identifying 13 motivations with 58 items [12]. To name a few, these motivations include understanding local culture when traveling, getting over ex-partners, passing time, and looking for various kinds of relationships. In exploratory factor analysis, the number of retained factors is only based on eigenvalues, and the parallel analysis method was not used. Notably, out of thirteen factors, there were five factors containing only three items. Despite the comprehensive nature of this scale, we chose to test and validate the work by Sumter et al. [5] because preliminary evidence has shown its applicability among young adults worldwide, rarely in Eastern society. Moreover, compared to the scale with thirteen motivations, the scale with six motivations is more parsimonious, allowing for efficient validation and assessment.

### 1.2. Methodological Issues of the Tinder Use Motivation Scale

Given the popularity of dating apps in mainland China, and that the Tinder use motivation scale has proved its utility among young users across the world, reexamination of its psychometric properties among Chinese young adults is highly warranted. In addition, there are other reasons to establish and reexamine the Chinese version of the motivation scale. 

First, in the work by Sumter et al. [5], the sample used for the development of the Tinder use motivation scale may be too small for modeling the multidimensional structure of the scale (i.e., n = 163). For instance, Costello and Osborne [13] cautioned that EFA is a “large sample” procedure, meaning that replication or generalization is unlikely if the sample is too small. Specifically, a larger subject-to-item ratio tends to produce more accurate solutions, with a 60% correction rate under larger ratios (10:1) and a 10% correction rate under smaller ratios (2:1). In the current case, the sample size should be at least 460 to reach a 60% correction rate. In addition, Comrey and Lee [14] suggested that an absolute sample size for an EFA analysis should be over 500. Furthermore, in the work of Sumter et al. [5], unraveling Tinder use motivation only conducted EFA based on CTT (classical test theory); however, confirmatory factor analysis is necessary for the validation of the scale because EFA is an error-prone procedure, even with large samples and optimal data [13]. As such, whether the scale has the same structure across a certain population should be validated with confirmatory factor analysis (CFA). To the best of our knowledge, CFA has not been conducted on the dating app use motivation scale in previous research.

Second, the original Tinder use motivation scale was developed and validated among Dutch emerging adults, and there may be cultural differences between Dutch culture and Chinese culture in the way that young adults from these two cultures perceive and respond differently to the items describing their preferences for looking for romantic or casual sex partners through dating apps. For instance, Chan indicates that Chinese dating app users strongly object to mudixing (i.e., purposefulness; referring to the direct, overt relationship-seeking practices that are prevalent on marriage websites and in matching by parents) [4]. Moreover, the original motivation scale was developed based on Tinder app use, but has been widely used across other dating apps [15]. Chinese users use a variety of dating apps, such as Momo and other counterparts, and it is necessary to examine the applicability of such a scale to other dating apps.

### 1.3. Modeling the Chinese Version of the Dating App Use Scale according to Item Response Theory (IRT)

More recently, powerful methods of conducting psychometric analyses, such as sample-free item response theory (IRT), have been encouraged, allowing for more reliable results to be obtained [16]. IRT evolved from CTT with the purpose of overcoming many of its shortcomings [17]. CTT is a traditional quantitative approach for testing the reliability and validity of a scale based on items, and assumes that each participant has a *true score*, which is not true. IRT attempts to explain the connection between recorded item responses on a scale and a latent construct. IRT also can estimate the probability of a specific response to an item. IRT offers many benefits relative to CTT, from sample-invariant parameter estimates (assuming no differential item functioning across populations) to metrics of reliability at both the test/scale and item levels that are conditional on the trait being assessed. According to IRT, items can be selected more reasonably and users can check the information for each item, which makes it possible to find the key items of a scale.

### 1.4. Objectives of the Present Research

Taken together, the current research aims to apply IRT and CTT frameworks to address the methodological issues in previous research, and examine the performance of the translated Tinder use motivation scale among Chinese emerging adults who are currently active dating app users. Our objective was to select qualified items from the original item pool based on IRT analysis, find the latent factors behind the manifest items, and then confirm them in a sample of Chinese adults under the CTT framework.

## 2. Methods

### 2.1. Sample and Procedures

In total, a sample of 1046 dating app users were recruited from the online sample provider Wenjuanxing in November 2019. From the total pool of preregistered users, Wenjuanxing recruited emerging adult dating app users (aged 18–30 years old) from all regions of mainland China. The invitation link to the survey was disseminated through email. Ethical approval for the study was granted by the corresponding author’s university. All participants were given notice that participation in this study would remain anonymous, and that all data collected were solely analyzed for scientific purposes. Prior to answering the questions, all participants signed consent forms. According to the answering setting, participants could not skip any question. There was no missing data in the present study.

To improve response quality, an attention-checking item was included in the survey. Respondents had to correctly answer the attention item (“Please choose ‘strongly disagree’ for this item”) to be included in the data analysis. Respondents who met these inclusion criteria, and also indicated informed consent to participate in the survey, received a small amount of monetary compensation for participation. Most participants reported using Tantan or Momo (Tinder’s Chinese counterpart) (71.8%).

### 2.2. Measures

#### 2.2.1. Demographics 

Participants reported their gender (0 = *male*; 1 = *female*, 52.30%), age (range: 18–30, *M* = 26.20, *SD* = 4.26), and sexual orientation (0 = *heterosexual*, 90.1%; 1 = *non-heterosexual* (homosexual, bisexual, or not sure). Furthermore, the highest education level (high school or less, 22.9%; undergraduates, 68.5%; graduates or above, 8.6%), monthly income (less than 12,500 per month, 93.8%), as well as marital status (single, 79.9%; married, 16.3%; others, 3.9%) were recorded.

#### 2.2.2. The Chinese Version of the Dating App Use Motivation Scale

The Chinese version of the scale was obtained by having two researchers conduct a translation of the complete 46-item pool of the Tinder use motivation scale (see Appendix A), presenting them in the same order. We used backward-translation techniques to translate the English scale into Mandarin. The two translators were native speakers of Mandarin and were also fluent in English. They first independently translated half of the items into Mandarin, discussed possible disagreement issues until these were resolved, and then translated the second half of the items. There was no disagreement in the translations. Although no specific cultural adaption had been applied to the items, special attention was paid to adjusting Chinese wording to contemporary Chinese and expressions reflective of young adults’ everyday lives. In regard to the response format, a Likert scale was used, with five options expressing levels of agreement (1 = *strongly disagree* to 5 = *strongly agree*). 

#### 2.2.3. Fear of Being Single as the Validity Criterion

Prior research suggests that users use dating apps to satisfy romantic needs, and dating apps make it very easy to meet others. Such technological affordance may re-emphasize the significance of being with someone in modern society, and experiencing single life may become particularly difficult for young adults [5]. Therefore, we assumed that fear of being single would be significantly correlated with relational-related motivations (i.e., seeking for relationship and ease of communication) and not with recreational motivations (i.e., fun and thrill of excitement). In fact, Timmermans and De Caluwé [12] reported that fear of being single was positively associated with using Tinder out of relationship seeking (*r* = 0.34, *p* < 0.01), flirting/social skills, and socializing motivation (*r* = 0.27, *p* < 0.01; *r* = 0.18, *p* < 0.05), suggesting fear of being single was an ideal validity criterion indicator for dating app use motivation scale. Fear of being single was included to determine convergent validity and measured with a 3-item scale (*M* = 2.61, *SD* = 0.94; Cronbach’s *α* = 0.69) [18].

### 2.3. Analytic Plan

Given the lack of knowledge on the factor structure of the dating app motivation measure among the Chinese population, CTT analyses, as well as IRT analyses, were carried out. We first used IRT to select qualified items from the original 46-item pool. Each item had five ordered response categories, which could be treated as ordered polytomous categories. For polytomous categorical responses, Masters’ partial credit model (PCM) and variants or Samejima’s graded response IRT model (GRM) were used. In comparison to PCM, the GRM fit the data reasonably well in most studies [19,20,21]. As the most commonly used model, GRM is a modern measurement method that overcomes some limitations of classical test theory methods [22], and was used for estimating item parameters in our study. In Samejima’s graded response model, the modal threshold of each option and the slope of each item are estimated. We also asked the software (eirt 2.0.3) to compute test information curves based on GRM, and showed the range at which the items provided the most information on latent traits (theta; θ). The objective of the first step of the analysis was to subject the same dating app motivation bank items to GRM, and we selected the better items to obtain a reasonable factor structure in the second step.

Second, using the selected items from the first step, we planned to explore the factor structure of dating app motivation by EFA; and third, applied confirmatory factor analyses (CFAs) using maximum likelihood estimation in different subsamples. The reasons why we did not choose to fit the latent structure obtained from previous study using the CFA model directly in the Chinese sample are: (1) the work by Sumter et al. [5] selected items under the CTT framework, which means the item parameters relied on the sample; it may include bias items, which performs well in a specific sample; (2) as we mentioned before, the sample size was small in Sumter et al. The indices we used to evaluate the model fit were the CFI (comparative fit index) and TLI (Tucker–Lewis index), the RMSEA (root mean square error of approximation), and the SRMR. CFI and TLI need to be greater than 0.90; both RMSEA and SRMR should be less than 0.08, indicating a good model fit [23,24]. Based on the best CFA model, we intended to check convergent validity in the same sample.

## 3. Results

### 3.1. Item Selection

The bank of items fulfilled the model assumptions and fitted the data reasonably well. Based on Samejima’s graded response IRT model [22,25], using the marginal maximum likelihood estimator (MMLE), we can obtain one slope and five threshold (location) parameters for every item. No item was ignored by the estimation process. The slope parameter is a measure of the discriminating power of an item, and the location parameter measures the frequency of a behavior or an attitude. The slopes and thresholds of all items were estimated and uploaded to figshare for readers to check (see, doi: 10.6084/m9.figshare.20024240. accessed on 8 June 2022). We took a slope larger than 1 and thresholds within [−4, 4] as the item selection standards [17,26]. Notably, we checked the item characteristic curve (ICC), also called the trace line, to visually check the overlap between neighboring categories [27]. If the adjacent categories had too much overlap, it seemed unreasonable to set them as current response options. Some response categories were merged or deleted, which diminished the final options. According to these standards, 21 items were excluded in the next step of the analysis. We picked 25 items in the CTT factor structure analysis (see Appendix B). For the item parameters (e.g., item difficulty and item-rest correlation) based on CTT, we estimated them and uploaded them to figshare for readers to check (see, doi: 10.6084/m9.figshare.21369834. accessed on 20 September 2022). Descriptive statistics of the remaining items based on IRT are presented in Table 1. 

According to the test information curves, information for all the selected items peaked at a wide range, as shown in Figure 1. We calculated all the items’ information and obtained the test information curve, which was translated into the reliability of the measure across the latent trait. To translate the amount of information into a standard error of estimation, we only needed to take a reciprocal of the square root of the amount of test information [25]. Information amount of 16 approximately equals an internal consistency of 0.937 [28]. In this way, the dating app motivation measure is reliable over a relatively wide range of latent traits for the present sample.

### 3.2. Exploratory Factor Analysis (EFA)

The free software *Jamovi* V2.2 [29], based on R, was used to run the EFA (n = 545, 51.4% female) and CFA (n = 501, 53.3% female), using the random half of the respondents in the development sample. The Kaiser-Meyer-Olkin (KMO) [30] measure of sampling adequacy, of which values range from 0.00 to 1.00, was used. KMO values larger than 0.70 are desired, indicating that the correlation matrix is factorable. In the present study, the results of Bartlett’s test of sphericity [31] indicated that the correlation matrix was not random; χ^2^ (300) = 4097, *p* < 0.001 and the KMO statistic [30] was 0.921, which is well above the minimum standard for conducting factor analysis. Therefore, it was determined that the correlation matrix was appropriate for factor analysis.

The scree plot of eigenvalues from the EFA (*Jamovi* uses the psych R package) in the developmental sample is shown in Figure 2. Through an examination of the scree plot, we derived one factor, as the leveling off clearly occurs after the first factor. Furthermore, a parallel analysis technique was used so that the obtained eigenvalues were compared to those that were obtained from random data. The number of meaningful factors was the number with eigenvalues greater than what would be found with random data. Parallel analysis (as shown in Figure 2) suggested that six factors should be retained. The total variance explained by the six factors was 41.9%. As with EFA, there were two main family approaches to rotation to obtain a better simple structure: orthogonal (e.g., Varimax) rotation assumes the extracted factors to be uncorrelated, whereas oblique (e.g., Oblimin) rotation allows the selected factors to be correlated. Practically, oblique solutions are arguably more sensible.

The six-factor solution was then examined for adequacy in the samples of the other half. The overall model fit indices are shown in Table 2. All the model fit measures showed that the six-factor solution fit the data very well. For example, a CFI value of 0.90 or greater [32] indicated an acceptable model fit. The factor loadings for each item are shown in Table 3. The loadings lower than 0.30 were hidden in this table.

The first motivation factor was “seeking a relationship”, reflecting the motivation to use dating apps to find a steady relationship. The second motivation factor was called “emotion-focused coping”, reflecting the motivation to use dating apps as a way of positively improving their emotional state, relaxing, or reducing loneliness. The third motivation factor was labeled “self-worth validation”, indicating the motivation to feel more confident and receive compliments about one’s appearance. The fourth motivation was identified as “the thrill of excitement”, referring to using dating apps to have novel, exciting, and exhilarating experiences. The fifth motivation was labeled “ease of communication”, reflecting the preference for online communication rather than offline. The last motivation was “fun”, referring to using dating apps for fun and pleasure.

The validity-criterion indicators showed meaningful correlations with these motivations. Specifically, the fear of being single was most strongly related to the motivation for seeking a relationship, indicating that higher levels of fearing being single were associated with greater motives to use dating apps for a romantic relationship (*r* = 0.233, *p* < 0.001). Self-esteem was most strongly related to the motivation for self-worth validation (*r* = 0.146, *p* < 0.001), and sensation-seeking was most strongly related to the motivation for the thrill of excitement (*r* = 0.196, *p* < 0.001). 

### 3.3. Confirmatory Factor Analysis (CFA)

CFA allows the researcher to test the hypothesis that a relationship exists between observed variables and their underlying latent constructs.

As shown in Table 4, fit measures in subsequent CFAs, such as CFI (0.931) and TLI (0.915) values, meet the criteria (0.90 or larger) for good model fit, confirming a six-factor latent structure for dating app motivation. Moreover, RMSEA indicates the amount of unexplained or residual variance. The 0.050 RMSEA value was smaller than 0.08, which also met the criteria (0.08 or less) for a good model fit. The SRMR (0.041) was also less than the recommended criterion of 0.08. All four fit statistics indicated an acceptable fit. The CFA analysis confirmed the factor structure from the EFA. Meanwhile, the factor loadings, as shown in Table 5, exceeded the thresholds for acceptable loadings, as they were all greater than 0.40. 

### 3.4. Reliability of Factors

The reliability indices (as shown in Table 6), which were measured by Cronbach’s alpha, factor loading-based McDonald’s omega, and mean intercorrelation (MIC), suggested that most of the items had a relatively acceptable internal consistency [33,34,35,36]. McDonald’s omega was an available option for replacing Cronbach’s alpha in some situations, requiring item loadings from a confirmatory factor analysis (CFA) [37]. Mean inter-item correlation is another way of analyzing internal consistency and reliability based on all possible paired correlations. The ideal range of mean inter-item correlation is from 0.15 to 0.50; less than 0.15 means that the items are not well correlated and do not measure the same construct. More than 0.50 means the items overlap too much or may be almost repetitive [38,39].

## 4. Discussion

The primary purpose of our study was to select qualified items to explore and confirm construct validity at the item level in the Chinese population under the IRT and CTT frameworks. Our study contributes to the literature in both methodological and theoretical aspects.

### 4.1. Addressing the Methodological Issues

First, the original Tinder use motivation scale was limited by using traditional CTT and a small sample. In the current research, we advanced the research by using IRT to select qualified items using a large sample. As previously noted, IRT has several pros and overcomes some cons of CTT. It is not a surprise that we had different selected items and explored different latent structures. Specifically, item selection procedures fell under several standards, and 21 items were potential candidates for deletion, leaving 25 items for the next step of analysis. Item selection based on information functions to match a target function has advantages over CTT item selection procedures [40]. For all 25 items, total item information, which is extremely useful in test design and evaluation, reached 16 at a wide range, meaning that it was reliable for most trait levels [25]. Although some items were dropped, the measurement precision was kept for different levels of participants. At the time of finishing the test, it was shorter than the full-length test. Exploratory factor analysis retracted six correlated factors based on evidence from the scree plot, eigenvalues, and parallel analysis.

As noted, the original Tinder motivation use motivation scale only conducted EFA. We further advanced the research by conducting CFA and consulted multiple determining indices to confirm the structure in an independent sample. Confirmatory factor analyses provided support for the six factors. In CFI and TLI, the incremental fit indices were below the commonly accepted cutoffs [32,41]. However, a common criticism of incremental fit indices is that they are reliant on the badness of fit of the null model [42]. While all correlations in a matrix are positive in a dataset, many of the item correlations are relatively low. Altogether, we viewed the results as generally supporting the six-factor structure of the dating app motivation scale without considering correlated errors.

The advanced CFA-based omega coefficients were adopted for each motivation’s reliability indicators. From this point, the present study made up for some methodological deficiencies in previous studies. For the measure of reliability, both coefficient alpha and coefficient omega were used in the present study [43,44,45]. Some subscales have low coefficient alphas such as ease of communication (0.546) and fun (0.660). Various reasons may cause Cronbach’s alpha to have a low value, such as a smaller number of items. The subscale named “fun” was the case. A low value for alpha may mean that there are not enough questions on the test. That is, it may explain the low coefficient alpha of the ease of communication subscale. In contrast, CFA-based reliability estimation-omega should be the new standard in reliability estimation [45]. In the present study, a coefficient omega of 0.671 represents the proportion of total-score variance that is due to the single factor of a subscale; that is, how reliably a total score for these three items measures the emotion-focused coping factor. For MIC, which provides an assessment of item redundancy, all values fell in the range of [0.15, 0.50], meaning that items on a factor assess the same content [38,39]. In the future, more reliability indices may be needed in this field [46].

Second, the present study validated the shared motivations that were reported in the original scale, identified unique motivations among Chinese adult users, and discussed the motivations that were not found in the current sample from the perspective of cultural differences and methodology concerns. Special attention was given to the sexual ideologies of traditional Chinese culture. In particular, our results demonstrated a different factor structure from that reported in the original scale, in which a six-factor structure was supported; however, four factors were basically duplicated in the original scale. Specifically, seeking a relationship (labeled “love” in the original scale), thrill of excitement, self-worth validation, and ease of communication were identified and validated in Chinese young adults. Notably, we revised the label “love” into “seeking a relationship”, and kept the labels for the other three motivations, as indicated in the original scale. The revision of the labeling was due to the cultural factor that traditional Chinese culture does not prioritize the concept of *ai* (love) in an intimate relationship [47]. Therefore, “to find a romantic partner” or “to find someone to be with” might better reflect the desire for a relationship than for love. 

The differences between the motivations found in the current results and those in the original scale also manifest in the following three ways. First, although dating apps have the reputation of encouraging users to initiate casual sex, and a handful of research has regarded such motivations as fundamental, the motivation for casual sex was not supported in the Chinese sample. A possible explanation is that the cultural difference between the two samples leads to the dissimilitude in the casual sex factor. In traditional Chinese society, sociosexual expression was considered irrational [48]. Even though Chinese ideology has evolved to recognize that sexuality has relaxed in the past several decades, the traditional avoidance of talking about sex may still exist, and having sex with causal partners may not be considered only a moral sin, but may also be harmful to health and life. Therefore, young Chinese people may not feel comfortable talking about sex or exchanging pictures. In fact, neither “finding a one-night stand” nor “finding someone to have sex with” were identified as primarily motivated behaviors on dating apps among young Chinese users. 

Second, the motivation of trendiness in the original scale was not supported in the current results. Instead, using dating apps just for fun and pleasure emerged as a main motivation. However, the “fun” motivation in the current results was related to trendiness, in the way that trendiness in the original scale was regarded as an entertainment need. One possible reason for the absence of the trendiness motivation may be that dating apps have been popular for several years, and researchers speculated that this motivation would become less strong over time when dating apps had become the established form of online dating [5]. Given that the number of monthly active users of Momo alone exceeds 20 million [3], we believe that this speculation was confirmed in the current sample. Third, the motivation for emotion-focused coping emerged as a unique factor that does not appear on the original scale. This motivation includes three items, reflecting that some young adults use dating apps mainly to positively improve their emotional state, relax, and reduce loneliness. 

### 4.2. Limitation and Future Research

The present findings should be interpreted with consideration of several limitations. Firstly, although we used a relatively large, heterogeneous sample of dating app users in China, adults older than 30 were underrepresented. However, females who were over 30 years old and had not married were called “leftover women” [49]. Such a discourse presents an increasing level of marriage pressure and anxiety among women over 30 years old. We expected female dating app users in this age group to use dating apps primarily for relationship-seeking motivations. Future research should use additional sampling methods to include a larger age span of dating app users in the sample. Secondly, our findings might not fully translate to dating app use in other countries due to cultural differences. For instance, self-worth validation motivation might be more salient among North American dating app users than Chinese users, as scholars argue that the need for high self-worth is greater in North America than in Eastern countries [50]. Future research should explore the cultural differences in specific motivations in further detail. Thirdly, the present study did not tie specific motivations to different dating apps. Different dating apps are likely designed to target audiences with different motivations. In fact, the “all-male” Grinder app is designed for male users who seek encounters with men [51]. Future research should link motivations with the types of dating apps among Chinese users, with particular attention paid to gay dating app use as the cultural prohibition against sexual minorities is strong in China [52].

## 5. Conclusions

The present study used IRT to improve the measurement of motivations referring to using dating apps. We deleted redundant, uninformative items from the original item pool to develop a Chinese version of the dating apps use motivation scale under the CTT and IRT frameworks. Finally, we retained the 21 most informative items. Findings supported the structural and convergent validity of the dating app use motivation scale in large samples. The removal of 25 uninformative items, without losing information, will improve the future efficiency of measurement of the dating app use motivation scale. The Chinese version of the dating apps use motivation scale could also function as a screening tool, and thus, facilitate the increased assessment of dating app use by practitioners and researchers in applied settings.

## Figures and Tables

**Figure 1 ijerph-19-13838-f001:**
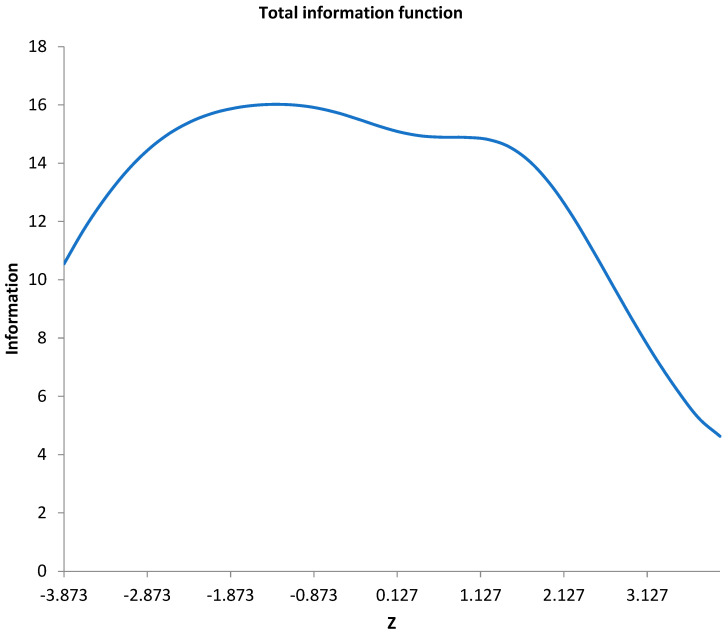
Test information curve.

**Figure 2 ijerph-19-13838-f002:**
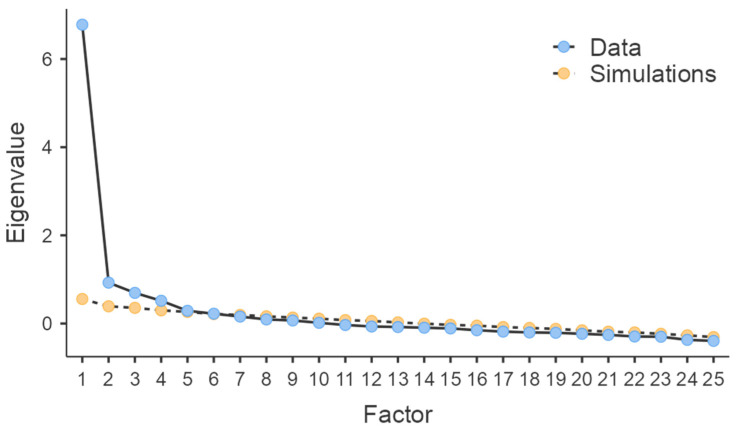
Scree plot of EFA.

**Table 1 ijerph-19-13838-t001:** Descriptive statistics of items.

Items	Slope	Threshold1	Threshold2	Threshold3	Threshold4	Threshold5
1	1.298	−3.761	−3.225	−1.896	0.315	1.743
2	1.554	−3.390	−2.543	−1.110	0.260	1.043
3	1.020	−4.045	−3.283	−1.942	−0.160	1.043
5	1.022	−2.823	−1.758	0.160	2.085	3.157
9	1.121	−2.531	−1.772	0.075	1.893	2.622
11	1.043	−2.799	−1.781	−0.139	1.304	2.122
12	1.491	−3.154	−2.420	−1.043	0.508	1.415
13	1.134	−3.854	−3.062	−1.670	0.159	1.388
14	1.038	−3.965	−3.106	−1.676	−0.047	1.009
19	1.630	−2.230	−1.575	−0.342	1.020	1.805
21	1.103	−2.514	−1.740	−0.416	1.131	2.127
22	1.175	−3.019	−2.216	−0.731	0.814	1.678
24	1.085	−3.886	−3.124	−1.679	0.258	1.511
26	1.377	−3.109	−2.361	−0.988	0.502	1.365
27	1.187	−2.885	−2.029	−0.482	1. 012	1.816
28	1.453	−3.467	−2.888	−1.617	0.102	1.128
34	1.076	−2.746	−1.861	−0.287	1.304	2.205
36	1.449	−2.285	−1.622	−0.424	0.865	1.619
37	1.139	−2.751	−2.113	−1.093	0.362	1.434
39	1.034	−3.232	−2.443	−0.966	0.924	2.128
40	1.595	−2.816	−2.204	−1.036	0.504	1.487
41	1.124	−1.906	−1.205	0.246	1.752	2.508
42	1.145	−3.581	−2.858	−1.599	0.255	1.572
43	1.309	−2.302	−1.637	−0.325	1.120	1.917
46	1.214	−2.509	−1.888	−0.579	1.285	2.462

**Table 2 ijerph-19-13838-t002:** Model fit measures.

	RMSEA 90% CI				Model Test		
RMSEA	Lower	Upper	TLI	BIC	χ²	*df*	*p*
0.0299	0.0217	0.0375	0.961	−794	246	165	<0.001

**Table 3 ijerph-19-13838-t003:** Factor loadings and inter correlations.

	Factor						Uniqueness
	1	2	3	4	5	6	
Q41	0.792						0.378
Q9	0.601						0.539
Q21	0.463					−0.259	0.556
Q46	0.443				0.207		0.598
Q12		0.693					0.481
Q13		0.647					0.588
Q14		0.455		−0.215			0.679
Q28		0.290				0.219	0.593
Q40			0.681				0.427
Q39			0.594				0.623
Q27			0.483				0.599
Q22			0.430		0.237	−0.230	0.578
Q42			0.317				0.756
Q26			0.213	0.234			0.655
Q11				0.766			0.414
Q34				0.659			0.543
Q19			0.245	0.437			0.520
Q43					0.299		0.686
Q24			0.208		0.216		0.734
Q37					0.560		0.661
Q36					0.540		0.531
Q5	0.250				0.427		0.656
Q1						0.477	0.527
Q2		0.258				0.337	0.510
Q3		0.247				0.293	0.693
F1							
F2	0.369						
F3	0.444	0.495					
F4	0.300	0.434	0.361				
F5	0.454	0.533	0.523	0.282			
F6	0.069	0.305	0.115	0.143	0.157		

Note: ‘Principal axis factoring’ extraction method was used in combination with a ‘oblimin’ rotation. F1 = Seek for Relationship; F2 = Emotion-focused Coping; F3 = Self-Worth Validation; F4 = Thrill of Excitement; F5 = Ease of Communication; F6 = Fun.

**Table 4 ijerph-19-13838-t004:** CFA fit measures.

				RMSEA 90% CI	
CFI	TLI	SRMR	RMSEA	Lower	Upper
0.931	0.915	0.0418	0.0505	0.0435	0.0574

**Table 5 ijerph-19-13838-t005:** CFA factor loadings.

Factor	Indicator	Estimate	SE	*Z*	*p*	Stand. Estimate
Factor 1	Q9	1.000 ^a^				0.609
	Q21	1.193	0.1159	10.29	<0.001	0.620
	Q41	1.366	0.1132	12.07	<0.001	0.747
	Q46	1.115	0.1005	11.09	<0.001	0.671
Factor 2	Q12	1.000 ^a^				0.727
	Q13	0.814	0.0743	10.95	<0.001	0.594
	Q14	0.867	0.0863	10.05	<0.001	0.546
Factor 3	Q22	1.000 ^a^				0.582
	Q27	1.123	0.1077	10.43	<0.001	0.620
	Q39	0.894	0.0970	9.21	<0.001	0.541
	Q40	1.045	0.0961	10.87	<0.001	0.690
	Q42	0.845	0.0887	9.53	<0.001	0.561
Factor 4	Q11	1.000 ^a^				0.632
	Q19	1.037	0.0898	11.55	<0.001	0.742
	Q34	0.833	0.0803	10.37	<0.001	0.569
Factor 5	Q5	1.000 ^a^				0.463
	Q36	1.558	0.1788	8.71	<0.001	0.638
	Q37	1.323	0.1665	7.95	<0.001	0.531
Factor 6	Q1	1.000 ^a^				0.669
	Q2	1.359	0.1153	11.79	<0.001	0.750

Note: ^a^ fixed parameter. F1 = Seek for Relationship; F2 = Emotion-focused Coping; F3 = Self-Worth Validation; F4 = Thrill of Excitement; F5 = Ease of Communication; F6 = Fun.

**Table 6 ijerph-19-13838-t006:** Reliability of each factor.

Factor	Item	Alpha	Omega	MIC
Seek for Relationship	9, 21, 41, 46	0.750	0.757	0.427
Emotion-focused Coping	12, 13, 14	0.664	0.671	0.393
Self-Worth Validation	22, 27, 39, 40, 42	0.732	0.737	0.349
Thrill of Excitement	11, 19, 34	0.694	0.698	0.441
Ease of Communication	5, 36, 37	0.546	0.567	0.316
Fun	1, 2	0.660	0.668	0.486

## Data Availability

The data that support the findings of this study are available upon request from the corresponding author. The data are not publicly available due to them containing information that could compromise research participant privacy/consent.

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
