# Peer review of "Modeling of the Chinese Dating App Use Motivation Scale According to Item Response Theory and Classical Test Theory"

_ijerph, 2022, doi:10.3390/ijerph192113838_

Round 1

Reviewer 1 Report

I'm very interested in the topic of this manuscript. But this manuscript needs to be majorly revised. In general, the structure of the manuscript is confusing, the level of analysis is confusing, and the results of the analysis do not support the authors' results very well. Some specific comments are as follows:

1. Instruction changed to "Introduction"

2. In "Introduction," Line27-49, the authors use a range of literature to discuss the positive and negative effects of using dating apps and to illustrate different perspectives on the role of dating apps. When I read Line50-Line56, I thought that the authors were trying to analyze the role of dating apps in the COVID19 pandemic, but the authors suggested that they were trying to address the question of "why do they use them". I think this needs to be revised to better show the research gaps and issues that the authors want to show.

3. First, 1.1 and 1.2 are more like literature review and theoretical background. The authors should have made it a separate section instead of an "introduction". Second, according to Line50-59, the authors want to address the question of "why use", but according to 1.1-1.3, the authors do not want to analyze what factors influence the use of dating apps, but to develop a scale. Therefore, it is recommended that the authors seriously consider the "Introduction" section to better present the issues and research gaps that the authors want to investigate.

4. In 2.1, how many questionnaires did you distribute and how many did you return? What is the return rate? How many valid questionnaires were returned? What is the effective rate?

5. In 2.2.2 I believe that the demographics are incomplete and suggest that the authors provide a table to show data on specific demographic characteristics.

6. I think the discussion of the analysis plan in 2.3 needs to be reorganized, with a brief and clear presentation of which methods were used and the order in which they were followed. I do not think it is necessary to present the thresholds for a certain indicator here (this part should be written in the results).

7. The authors propose a model, but what about the model and its hypotheses? I feel that the author is very vague in this part. Please provide the equation or diagram of the model. If there are hypotheses, the development of the hypotheses needs to be written in detail.

8. In 3.3. The authors raised the hypothesis and latent variables, so where did the authors hypothesis development written? And only by CFA does not verify the relationship between latent variables.

9. The discussion section suggests a reframing around the research questions.

Author Response

We really appreciate your comments and suggestions. For the point-to-point response, please see the attachment.

Reviewer 2 Report

1) Please add a new Theoretical Foundation section to better explain: (i) Item Response Theory and (ii) Classical Test Theory 2) Section 2.2.1:   Expand the explanation of the data in Section 2.2.1 (Demographics). 3) Improve the quality of Figure 2. 4) Include most recent references from the last three years: 2020 to 2022.

Author Response

We really appreciate your comments and valuable suggestions. For the point-to-point response, please see the attachment.

Round 2

Reviewer 1 Report

The author addressed the comments I made very well.I think it's ready to be published.

Author Response

Thank you very much for your comments and insightful suggestions.